# GC Content in Nuclear-Encoded Genes and Effective Number of Codons (ENC) Are Positively Correlated in AT-Rich Species and Negatively Correlated in GC-Rich Species

**DOI:** 10.3390/genes16040432

**Published:** 2025-04-05

**Authors:** Douglas M. Ruden

**Affiliations:** C. S. Mott Center for Human Growth and Development, Institute for Environmental Health Sciences, Department of Obstetrics and Gynecology, Wayne State University, Detroit, MI 48201, USA; douglasr@wayne.edu; Tel.: +1-313-577-6688

**Keywords:** GC content, effective number of codons (ENC), codon bias, bimodal distributions, unimodal distributions, two-rank order normalization (TRON), CpG DNA methylation, epitranscriptomics

## Abstract

Background/Objectives: Codon usage bias affects gene expression and translation efficiency across species. The effective number of codons (ENC) and GC content influence codon preference, often displaying unimodal or bimodal distributions. This study investigates the correlation between ENC and GC rankings across species and how their relationship affects codon usage distributions. Methods: I analyzed nuclear-encoded genes from 17 species representing six kingdoms: one bacteria (*Escherichia coli*), three fungi (*Saccharomyces cerevisiae*, *Neurospora crassa*, and *Schizosaccharomyces pombe*), one archaea (*Methanococcus aeolicus*), three protists (*Rickettsia hoogstraalii*, *Dictyostelium discoideum*, and *Plasmodium falciparum*),), three plants (*Musa acuminata*, *Oryza sativa*, and *Arabidopsis thaliana*), and six animals (*Anopheles gambiae*, *Apis mellifera*, *Polistes canadensis*, *Mus musculus*, *Homo sapiens*, and *Takifugu rubripes*). Genes in all 17 species were ranked by GC content and ENC, and correlations were assessed. I examined how adding or subtracting these rankings influenced their overall distribution in a new method that I call Two-Rank Order Normalization or TRON. The equation, TRON = SUM(ABS((GC rank_1_:GC rank_N_) − (ENC rank_1_:ENC rank_N_))/(N^2^/3), where (GC rank_1_:GC rank_N_) is a rank-order series of GC rank, (ENC rank_1_:ENC rank_N_) is a rank-order series ENC rank, sorted by the rank-order series GC rank. The denominator of TRON, N^2^/3, is the normalization factor because it is the expected value of the sum of the absolute value of GC rank–ENC rank for all genes if GC rank and ENC rank are not correlated. Results: ENC and GC rankings are positively correlated (i.e., ENC increases as GC increases) in AT-rich species such as honeybees (R^2^ = 0.60, slope = 0.78) and wasps (R^2^ = 0.52, slope = 0.72) and negatively correlated (i.e., ENC decreases as GC increases) in GC-rich species such as humans (R^2^ = 0.38, slope = −0.61) and rice (R^2^ = 0.59, slope = −0.77). Second, the GC rank–ENC rank distributions change from unimodal to bimodal as GC content increases in the 17 species. Third, the GC rank+ENC rank distributions change from bimodal to unimodal as GC content increases in the 17 species. Fourth, the slopes of the correlations (GC versus ENC) in all 17 species are negatively correlated with TRON (R^2^ = 0.98) (see Graphic Abstract). Conclusions: The correlation between ENC rank and GC rank differs among species, shaping codon usage distributions in opposite ways depending on whether a species’ nuclear-encoded genes are AT-rich or GC-rich. Understanding these patterns might provide insights into translation efficiency, epigenetics mediated by CpG DNA methylation, epitranscriptomics of RNA modifications, RNA secondary structures, evolutionary pressures, and potential applications in genetic engineering and biotechnology.

## 1. Introduction

Codon usage bias, the non-random use of synonymous codons encoding the same amino acid, plays a crucial role in gene expression and genome evolution [1]. The effective number of codons (ENC) is a widely used measure of codon usage bias, ranging from 20 (where each amino acid in a gene is encoded by a single codon) to 61 (where all synonymous codons in a gene are used at least one time) [2,3]. Many factors influence codon bias, including mutational pressure, translational efficiency, and selection for optimal tRNA usage [4,5,6]. Similarly, GC content, defined as the proportion of guanine and cytosine nucleotides in each nuclear-encoded gene coding sequence, is an important genomic feature affecting gene regulation, DNA stability, and evolutionary adaptation [7,8,9,10,11]. DNA methylation at CpG sites in both promoter and enhancer regions and the coding region can regulate transcriptional regulation and alternative mRNA splicing [12,13,14,15,16,17,18,19]. Previous studies have shown that both ENC and GC content exhibit characteristic distributions across species, often following unimodal or bimodal patterns [20,21,22,23,24,25]. However, the relationship between ENC and GC content has been less extensively explored, and its evolutionary and functional implications remain unclear.

Studies on various organisms have reported mixed findings regarding the correlation between ENC and GC content. In some species, ENC and GC content are positively correlated, suggesting that increased GC content promotes codon diversity, possibly due to mutational biases or selection for GC-rich codons in highly expressed genes [26,27,28]. In contrast, other species exhibit a negative correlation between ENC and GC content, implying that highly biased codon usage occurs in GC-rich genes, potentially due to translational selection favoring specific tRNA pools [29,30]. These opposing trends raise fundamental questions about the evolutionary forces shaping codon usage and GC content across different taxa.

Two-Rank Order Normalization (TRON) is a new mathematical method I developed to compare lists of items with different types of characteristics to determine whether the characteristics are correlated. For example, if you have 100 people, you can rank them in terms of height and hair color and see these traits are correlated. One might hypothesize, for instance, that taller people tend to have lighter hair, and this can be tested by the TRON method. In this paper, I performed TRON calculations for two of the many characteristics of nuclear-encoded genes, namely GC content and ENC levels. The number of nuclear-encoded genes ranges from several thousand in prokaryotes to several tens of thousands in eukaryotes. One advantage of the TRON method over other correlation methods is that the TRON method provides insights into the two-dimensional relationships of the correlations, such as the bimodal GC and ENC distributions. 

Characteristics of each gene, such as GC content or ENC levels, are available on tables such as in the Codon Statistics Database [31]. Before I began these analyses, I predicted that, as GC content increases, ENC levels for these genes would decrease because there are fewer As and Ts available to make codons. As I describe in this paper, my prediction was true for species with high GC content, such as rice (*Oryza sativa*), mice (*Mus musculus*), and humans (*Homo sapiens*), but, surprisingly, the opposite is true for species with low GC (i.e., high AT) content, such as bees (*Apis mellifera*) and other species, where ENC levels decrease as GC content increases.

In this study, I systematically analyze ENC and GC rank distributions across 17 species with a range of GC content in nuclear-encoded genes. I find that ENC rank and GC rank exhibit species-specific correlations, i.e., a positive correlation in bees and a negative correlation in rice. Moreover, I demonstrate, using the TRON method, that the mathematical interplay between these distributions leads to opposite outcomes—subtraction of ENC from GC (GC-ENC) rank distributions results in a unimodal distribution pattern in bees, and a bimodal distribution pattern in rice. Also, the addition of GC rank and ENC rank (GC+ENC) produces a bimodal distribution pattern in bees and a unimodal distribution pattern in rice. These findings provide new insights into the complex interactions between codon usage and genomic composition, with potential implications for understanding translation efficiency, genome evolution, and species-specific selective pressures.

My results contribute to the broader field of comparative genomics by elucidating how different species maintain distinct codon usage strategies and GC content distributions. I discuss possible evolutionary mechanisms driving these correlations, including mutational biases, selection for translational efficiency, and constraints imposed by tRNA availability. Finally, I explore the practical applications of understanding ENC and GC relationships in evolution, as well as in optimizing gene expression in synthetic biology and biotechnology [32,33,34,35].

## 2. Materials and Methods

All the GC (G or C content for a nuclear-encoded gene) and ENC (effective number of codons for a nuclear-encoded gene) data used in this paper are from the Codon Statistics Database (http://codonstatsdb.unr.edu, accessed on 15 March 2025) [31]. I performed all of analyses in this paper on Excel™ (version 16.89.1).

To generate the rank order for ENC, the “gene stats” table for a particular species (e.g., the mouse, *Mus musculus*) was downloaded into an Excel™ file and sorted (low to high number) by ENC (with a lowest possible value of 20 and a highest possible value of 61) and the genes were numbered in the order of 1, 2, 3, … N_ENC_, where N_ENC_ is a nuclear-encoded gene sorted by ENC value. Genes that had an NA (not applicable) rating for an ENC value at the end of the table were deleted, presumably because these genes do not encode all 20 amino acids, but these were generally fewer than 10–50 genes.

To generate the rank order for GC, while keeping the rank order for ENC, the entire Excel™ table was next sorted (low to high number) by GC content (with a lowest possible value of 0.00 and a highest possible value of 1.00), after removing the ENC NA rows, and the genes were numbered 1, 2, 3, … N_GC_, where N_GC_ is a nuclear-encoded gene sorted by GC content.

To generate correlations between GC rank and ENC values, the entire Excel™ table was sorted by GC rank and a scatter plot of the ENC values (y-axis) vs. GC rank (x-axis) was plotted using Excel™ with the “insert chart > scatter plot” function. On the scatter plots, GC rank is the x-axis and ENC value is the y-axis (Figure 1). Trendlines were added by right clicking a point on the scatter plot and clicking “add trendline.” Next, “set intercept” was checked, “y-intercept” was added (INTERCEPT = (GC rank column, ENC rank column), “display equation on chart” was checked, and “display R^2^ value on chart” was checked.

To generate the GC-ENC and GC+ENC histograms, N_GC_-N_ENC_ and N_GC_+N_ENC_ columns were selected for each species and plotted using Excel™ with the “insert chart > histogram” function.

To perform Two-Rank Order Normalization (TRON) analyses, I used the equation: TRON = SUM(ABS((A_1_:A_N_) − (B_1_:B_N_))/SUM(ABS(A_1_:A_N_) − (R_1_:R_N_)), where A_1_:A_N_ is a rank-order series of one trait (e.g., GC), B_1_:B_N_ is a rank-order series of a second trait (e.g., ENC), sorted by the rank-order series of the first trait. For example, if a gene has the lowest GC rank (A = 1) and the 10th lowest ENC rank (B = 10), then the first number in the ((A_1_:A_N_) − (B_1_ − B_N_)) series is −9 (i.e., A–B = −9). The series R_1_-R_N_ is a randomization of the number of items that are ranked (e.g., genes) using the equation SORTBY(A_1_:A_N_, RANDARRAY(N)). RANDARRAY differs from RAND in that the former uses each number in an array only once, whereas RAND chooses a random number within a range (e.g., 1–1000) for each number in the array.

## 3. Results

### 3.1. Description of GC and ENC Histograms in Bees (Apis mellifera), Rice (Oryza sativa), and Yeast (Saccharomyces cerevisiae)

GC histograms of nuclear-encoded genes are shown for bees (blue), rice (red), and yeast (green) (Figure 1a). They were plotted overlapping on the same graph with Adobe Photoshop™, using 100% opacity for layer 1 (bee) and 50% opacity for layers 2 (rice) and 3 (wasp). These histograms demonstrate that there is a wide range of GC content from very low GC (the lowest is 20% GC) in bees and very high GC (the highest is 81%) in rice. Also, both bees and rice, as well as many other insects and plants, have bimodal GC peaks, as was demonstrated earlier [20,21,22,23,24,25] (see also Figure 2 and Figure 3). Yeast has a unimodal GC peak approximately in the center of the histogram (Figure 1a, green). In addition to bees, rice, and yeast, I analyzed a total of 17 species for this paper (Figure 2, Figure 3, Figure 4 and Figure 5).


Figure 1Bees (*Apis mellifera*), rice (*Oryza sativa*), and yeast (*Saccharomyces cerevisiae*) have different patterns of GC content and ENC ranks for nuclear-encoded genes. Figure 2d–f shows the three histograms separately for clarity. (**a**) Bees (blue), rice (red), and yeast (green) GC content (0.00 to 1.00) histograms of all nuclear-encoded genes (x-axis) are plotted against the number of genes with that range of GC content (y-axis). (**b**) Bees, rice, and yeast ENC level (20–61) histograms of all nuclear-encoded genes (x-axis) are plotted against the number of genes with that range of GC content (y-axis). The overlaps of the histograms are shown in different shades, as indicated.
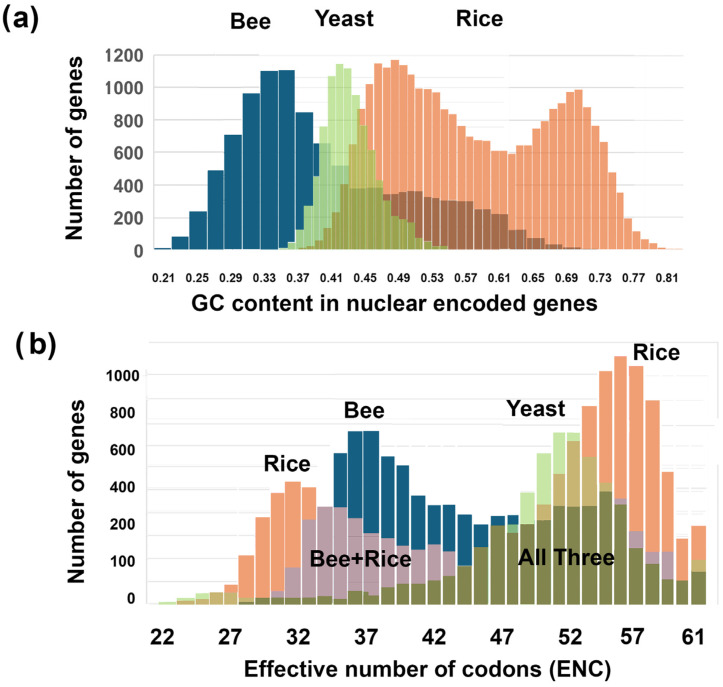




Figure 2First column, bee (*Apis mellifera*), second column, rice (*Oryza sativa*), and third column, yeast (*Saccharomyces cerevisiae*) correlations between GC content and ENC level for nuclear-encoded genes. (**a**–**c**) Correlations between ENC ranks (y-axis) and GC ranks (x-axis) for bees, rice, and yeast. ENC rank was determined by sorting all columns based on ENC levels (20–61) and then numbering the rows 1-N, where N is the number of genes in that species. GC rank was determined by sorting all columns based on GC levels (0.00–1.00) and then numbering the rows 1-N. When GC levels are sorted and all columns are selected, the original ranks of the ENC levels are maintained. Correlations between ENC levels and GC levels by selecting the ENC ranked column and making a scatter plot (shown in blue bars). Trend lines were made by right-clicking (control clicking) a point on the graph and selecting TRENDLINES (red arrows). Under TRENDLINES, select boxes for set intercept (=INTERCEPT(GCrank:ENCrank)), display equation on chart, and display R-squared value on chart (shown). Notice that bees have a positive correlation, rice has a negative correlation, and yeast has no correlation between ENC rank and GC rank (red arrows). (**d**–**f**) GC histograms for bees, rice, and yeast. The GC contents (0–1.00) for all nuclear-encoded genes are on the x-axis and the number of genes with that range of GC values is on the y-axis. Histograms were made by selecting the GC column and selecting histogram chart under the INSERT tab. Notice that bees and rice have bimodal distributions of GC content and Yeast has a unimodal distribution. (**g**–**i**) ENC histograms for bees, rice, and yeast. The ENC levels (20–61) for all nuclear-encoded genes are on the x-axis and the number of genes with that range of ENC levels are on the y-axis. Notice that bees and rice have bimodal distributions of ENC and Yeast has a unimodal distribution. (**j**–**l**) GC rank minus ENC rank histograms for bees, rice, and yeast. Notice that GC rank minus ENC rank (GC-ENC) is unimodal in bees and bimodal in rice. (**m**–**o**) GC rank plus ENC rank histograms for bees, rice, and yeast. Notice that GC+ENC is bimodal in bees and unimodal in rice. This is the opposite of the pattern in (**j**–**l**).
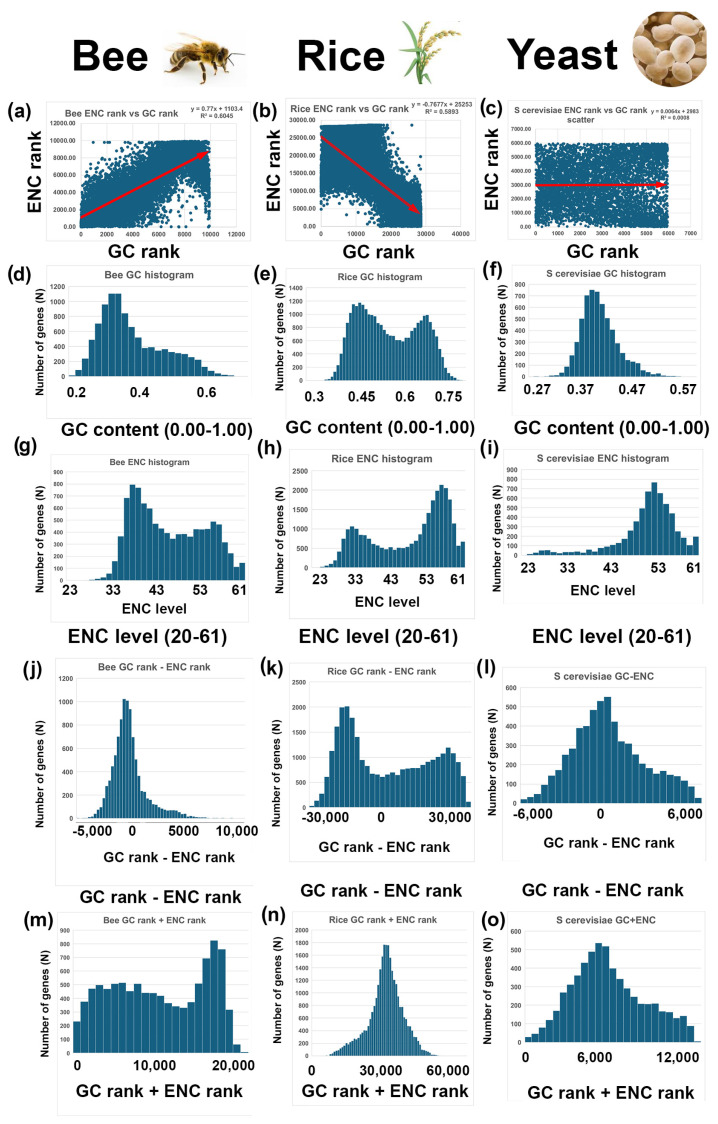

Figure 3GC and ENC analyses with negative correlations between GC rank and ENC rank: Mosquito (*Anopheles gambiae*), pufferfish (*Takifugu rubripes*), human (*Homo sapiens*), bread mold (*Neurospora crass*), banana (*Musa acuminata*), and mouse (*Mus musculus*). (**a**) Mosquito GC rank (y-axis) versus ENC rank (x-axis) shows a negative correlation. X-axis is 1 to 12,402 for the rank order of the 12,402 nuclear encoded mosquito genes based on GC content (0.00 to 1.00). Y-axis is 1 t0 12,402 for the rank order of genes based on GC levels, sorted on ENC rank (see Figure 2). (**b**) Mosquito histogram of GC content (0 to 1.00) versus the number of genes (N) that fall within the indicated range of GC content. (**c**) Mosquito histogram of ENC levels (0 to 1.00) versus the number of genes (N) that fall within the indicated range of ENC levels. (**d**) Mosquito histogram of GC rank—ENC rank versus the number of genes (N) that fall within the indicated range of GC rank—ENC rank. The x-axis is −12,402 to +12,402. (**e**) Mosquito histogram of GC rank + ENC rank versus the number of genes (N) that fall within the indicated range of GC rank + ENC rank. The x-axis is 1 to 2 × 1204, which is two times the number of nuclear-encoded genes in mosquitoes. (**f**–**j**) Pufferfish analyses (as described in (**a**–**e**)) for the 22,104 nuclear-encoded genes in this species. (**k**–**o**) Human analyses (as described in (**a**–**e**)) for the 19,708 nuclear-encoded genes in this species. (**p**–**t**) Bread mold analyses (as described in (**a**–**e**)) for the 9728 nuclear-encoded genes in this species. (**u**–**y**) Banana analyses (as described in (**a**–**e**)) for the 30,700 nuclear-encoded genes in this species. (**z**–**dd**) Mouse analyses (as described in (**a**–**e**)) for the 22,405 nuclear-encoded genes in this species.
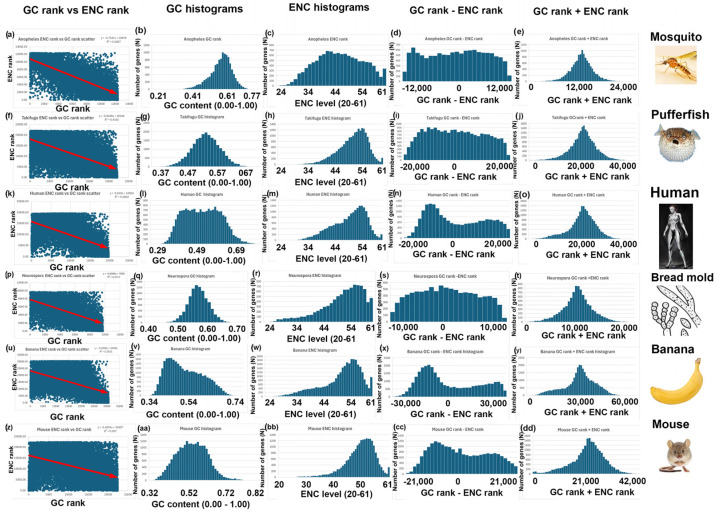

Figure 4GC and ENC analyses of species with positive correlations between GC rank and ENC rank: wasp (*Polistes canadensis*), rickettsia (*Rickettsia hoogstraalii*), slime mold (*Dictyostelium discoideum*), arabidopsis (*Arabidopsis thaliana*), and plasmodium (*Plasmodium falciparum*). (**a**–**e**) Wasp analyses (as described in Figure 4) for the 9854 nuclear-encoded genes in this species. (**f**–**j**) Rickettsia analyses (as described in Figure 4) for the 1663 nuclear-encoded genes in this species. (**k**–**o**) Slime mold analyses (as described in Figure 4) for the 13,078 nuclear-encoded genes in this species. (**p**–**u**) Arabidopsis analyses (as described in Figure 4) for the 10,160 nuclear-encoded genes in this species. (**v**–**y**) Plasmodium analyses (as described in Figure 4) for the 5321 nuclear-encoded genes in this species.
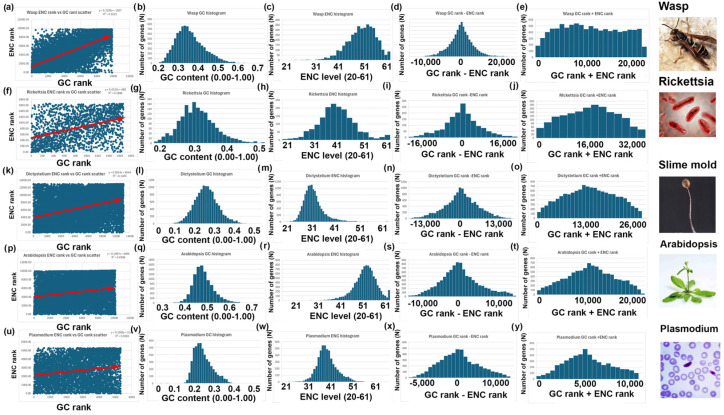

Figure 5GC and ENC analyses of species with little or no correlations between GC rank and ENC rank: *E. coli* (*Escherichia coli*), pombe (*Schizosaccharomyces cerevisiae*), and methanobacteria (*Methanococcus aeolicus*). (**a**–**e**) *E. coli* analyses (as described in Figure 4) for the 10,276 nuclear-encoded genes in this species. (**f**–**j**) Pombe analyses (as described in Figure 4) for the 5110 nuclear-encoded genes in this species. (**k**–**o**) Methobacteria analyses (as described in Figure 4) for the 1485 nuclear-encoded genes in this species.
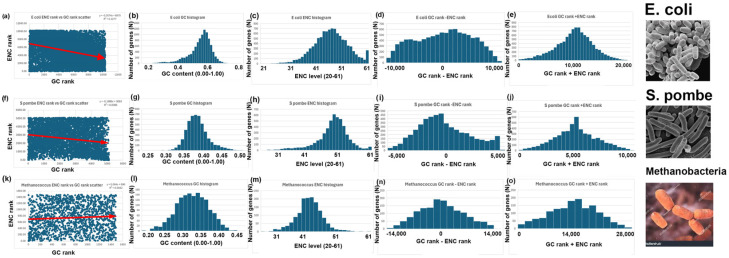



ENC histograms for nuclear-encoded genes are shown overlapping for bees (blue), rice (red), and yeast (green) (Figure 1b). As with the GC histograms, the ENC histograms are bimodal for both bees and rice and unimodal in yeast. ENC levels vary from very low (the lowest is 20, which means that each amino acid in that gene has only one codon) to very high (there are sixty-one codons for amino acids and three stop codons).

### 3.2. Description of Two-Rank Order Normalization (TRON) Mathematics

To describe this method, I will start with some definitions (Figure 6. Line A (number series 1, 2, …, 1000) and Line B (number series 1000, 999, …, 1) are graphed (blue and red lines, respectively, Figure 6a). The sum of A and the sum of B can be determined with the “Nth triangle number formula” (N(N + 1)/2) (Figure 6a). This equation is so named because of the triangular shape that is made [36]. This equation is also called “Gauss’ trick” because, according to lore, in the 1780s, a German schoolteacher assigned his class the task of summing the first 100 integers, expecting it to take a while. However, young Carl Friedrich Gauss quickly found the answer—5050—by recognizing a pattern: pairing numbers from opposite ends of the sequence (1 + 100, 2 + 99, etc.), each summing to 101, with 50 such pairs yielding 50 × 101. This insight led to the general formula for summing consecutive numbers: N(N + 1)/2 [36].

Line A+B (number series 1001, 1001, …, 1001) and Line A-B (number series −999, −997, …, 997, 999) are graphed (blue and red lines, respectively) (Figure 2b). The sum of A+B is N(N+1) (i.e., 2(N(N + 1)/2) and the sum of A-B, using the Excel™ function SUM(ABS(series A- series B)), is 500,000, which is exactly (N^2^/2) (Figure 6b).

Next, I generated a randomized list of Line A (Random) with Excel with SORTBY and RANDARRAY functions (i.e., SORTBY(A1:A1000,RANDARRAY(1000)). A+Random shows a triangular distribution (Figure 6c). Similary A-Random, graphed with the histogram chart, also shows a distribution that resembles a triangular distribution (Figure 6d). The RANDARRAY function differs from the RAND function because the former uses each number in the column only once, whereas the RAND function uses a random number in the range for each row (Excel™ tutorials).

This was repeated with Line A’ (number series 1, 2, …, 10,000) and Random’ was made with all 10,000 numbers represented once in the column. A’+Random’ shows a triangular distribution (Figure 6e). Similarly, A’-Random’ shows a triangular-like distribution that is much more clearly triangle shaped that A-Random (1–1000) (Figure 6f).

To calculate the sums of A+Random and A’+Random’, I added the numbers in each column and found that A+Random = 500,500 and A’+Random’ = 50,005,000. In my example, it does not matter whether the numbers are in order or randomized when they are summed because each number is present only once with the RANDARRAY function, and addition does not lead to the loss of any numbers. The solution is twice the Nth triangular number formula (i.e., N(N + 1)) because the series 1, 2, …, 1000 and the random array are added together in A+Random, thus doubling the number. In summary, (A + R) = N(N + 1), exactly, in all cases (Figure 6e)

To calculate the areas of A-Random and A’-Random’, I summed the numbers in each column, using ABS to generate absolute values since approximately half of the numbers are negative. I found that A-Random = 326,926 and A’-Random’ = 33,249,974. I noticed that the solutions are approximately N^2^/3, which would be 333,333 and 33,333,333, for A-Random and A’-Random, resepectively (Figure 6f. This makes logical sense because the range of A-Random would be from 0.00 (Line A—Line A, where both are series 1, 2, …1000) to 500,000 (i.e., N^2^/2). One hundred permutations of A-Random (i.e., (A-R_1_ + A-R_2_, … + A-R_100_/100) is approximately (N^2^)/3). (Figure 6g,h). I did a mathematical proof that indicates that the exact solution for E[S], the expected number for distribution S, is exactly N(N + 1)(2N−2)/6N. For large N, then the answer is approximately N^2^/3, as I estimated with my permutation analysis.

### 3.3. Comparisons Between GC and ENC Distributions

GC and ENC distributions are positively correlated in bees (*Apis mellifera*) and negatively correlated in rice (*Oryza sativa*). There is little or no correlation between GC and ENC in yeast (*Saccharomyces cerevisiae*).

Bees display a positive correlation between GC rank and ENC rank, meaning that as GC rank increases, ENC rank also increases (Figure 2a, red arrow; R^2^ = 0.60, Table 1). This indicates that genes with higher GC content tend to use a broader range of codons, while genes with lower GC content use a more restricted set. This might be partly explained by the lower average GC content in bees compared to rice, especially in bee GC peak 1 (Figure 2d).

In contrast to bees, in rice, which has a high GC content in nuclear-encoded genes, GC rank and the effective number of codons (ENC) exhibit a negative correlation, meaning that as GC content increases, ENC decreasess (Figure 2b; red arrow). This finding suggests that genes with higher GC rank tend to use a more restricted set of codons (lower ENC), possibly due to selection for specific tRNA availability or translational efficiency. Conversely, genes with lower GC rank use a more diverse range of codons (higher ENC), potentially indicating relaxed selection on codon usage bias. The positive correlation between GC and ENC in bees suggests a different evolutionary strategy in bees compared to rice, possibly reflecting differences in genome organization, translational selection, or adaptation to distinct environmental pressures.

There is no correlation between GC rank and ENC rank in yeast (Figure 3c). Also, yeast has unimodal GC and ENC distributions (Figure 2f,i). Yeast also has unimodal GC-ENC rank and GC+ENC rank distributions (Figure 2l,o).

2.Rice (*Oryza sativa*) has bimodal GC and ENC distributions, GC-ENC is a bimodal distribution, and GC+ENC is a unimodal distribution; (2 − 2 = 2; 2 + 2 = 1).

In rice, both GC content and the effective number of codons (ENC) follow a bimodal distribution (Figure 2e,h). However, rice GC+ENC is a unimodal distribution (Figure 2n). If GC rank and ENC rank were not correlated, a unimodal distribution is expected for GC rank + ENC rank based on the Two-Rank Order Normalization (TRON) method described above. However, I found that the sum of GC-ENC (i.e., SUM(ABS(GC-ENC)) in rice is 48% larger than expected by the equation: SUM(ABS)(A-R)/((N^2^)/3) (Figure 6e; Table 1, second to last column). In other words, even though rice GC-ENC historgram resembles a triangle distribution (i.e., unimodal), the sum of all the GC-ENC rows for the 28,571 nuclear-encoded genes in rice is 1.48 times the expected value if GC ranks and ENC ranks were randomly associated (Table 1, second to last column). I interpret this as further evidence that GC ranks and ENC ranks are highly correlated.

Rice GC+ENC is a bimodal distribution. However, SUM(GC+ENC) = SUM(A + R) = N(N + 1), and SUM(GC + ENC)/SUM(A + R) = N(N + 1)/N(N + 1) = 1.00 for all 17 species (Table 1, last column). Because of this limitation, Two-Rank Order Normalization (TRON) can only be done when subtracting two series of ranks, such as GC-ENC, and not when adding two series of ranks, such as GC+ENC. Therefore, for this paper, I will focus only on (GC-ENC) normalization to SUM(ABS(A-R)) = ((N^2^)/3) in this paper (Table 1, second to last column).

As a shortcut, I refer to rice having a (2 − 2 = 2; 2 + 2 = 1) pattern, which means that the GC distribution is bimodal and the ENC distribution is bimodal, GC-ENC is bimodal, and GC+ENC is unimodal. One possible explanation is that different functional classes of genes contribute differently to the ENC and GC distributions. While their individual distributions appear continuous, the way they interact—such as certain genes having disproportionately high or low ENC relative to their GC content—creates two separate peaks when subtracted. This may reflect distinct evolutionary pressures acting on different subsets of genes, such as differences in selection for codon usage bias, gene expression levels, or functional constraints.

3.The bee (*Apis mellifera*) has bimodal GC and ENC distributions, GC-ENC has a unimodal distribution, and GC+ENC forms a bimodal distribution (2 − 2 = 1; 2 + 2 = 2)

In bees, both GC content and the effective number of codons (ENC) exhibit bimodal distributions, meaning that each measure clusters into two distinct peaks rather than forming a single continuous distribution (Figure 2m). As expected for random distributions, GC-ENC forms a unimodal peak. However, unexpectedly, GC+ENC forms a bimodal distribution (Figure 3m). Given that both GC and ENC are bimodal, one might expect that adding ENC and GC (GC+ENC) would result in a unimodal distribution. This is because A+Random forms a unimodal triangle distribution (Figure 6f). This result (i.e., GC+ENC is bimodal) suggests that there are underlying relationships between these two variables (GC and ENC) that maintain a bimodal distribution when added.

As a shortcut, I refer to bees having a (2 − 2 = 1; 2 + 2 = 2) pattern, which means that the GC distribution is bimodal and the ENC distribution is bimodal, and GC-ENC is unimodal and GC+ENC is bimodal. The 2 + 2 = 2 pattern could indicate a balancing effect of evolutionary pressures, such as codon usage adaptation and GC content constraints, leading to a more uniform distribution when considering their difference. Such a pattern might reflect underlying genomic organization or selection pressures that maintain a more stable relationship between GC content and codon usage across the genome.

4.I repeated the analyses described above for bees, rice, and yeast with 14 other species. Six additional species have a negative correlation between GC and ENC, as I found with bees (Figure 3). Five additional species have a positive correlation between GC and ENC, as I found with rice (Figure 4). Three additional species have little or no correlation between GC and ENC, as I found with yeast (Figure 5).5.Two-Rank Order Normalization (TRON) was plotted for the 17 species analyzed in this study (SUM(ABS(GC-ENC))/(N^2^/3)). Table 1 is a summary of several of the variables that were used to make Figure 7. I compared all the variables with each other and highlighted the strongest correlations in Figure 7. I found a strong inverse correlation between TRON in the 17 species with slope of the GC and ENC correlations in the 17 species (Figure 7a; R^2^ = 0.98). I also found a strong second-order parabolic (ax^2^-bx) correlation between R^2^ and the slope of the GC versus ENC correlations (Figure 7b: R^2^ = 0.99).

I found a good correlation between the GC content in peak 1 in the 17 species (R^2^ = 0.24) and a better correlation between the GC content in peak 2 in the 17 species (Figure 7c; R^2^ = 0.60). Since two peaks can be mostly overlapping, such has the GC peaks in mice, if there was a unimodal GC distribution, I set peak 1 = peak 2 (Table 1). This is an interesting result, but I consider it tentative because of the limited number of species that I analyzed for this paper (17). This result could be influenced by selection bias, for instance, because I purposely chose species with known bimodal GC content distributions. Further analyses with a much larger number of species is needed to validate this result.

## 4. Discussion

### 4.1. Interpretation of GC and ENC Distributions Across Species

My findings reveal distinct patterns in the relationships between GC content and the effective number of codons (ENC) across different species, demonstrating unexpected correlations that challenge prior assumptions. In rice, both GC content and ENC exhibit bimodal distributions, and their difference (GC-ENC) forms a bimodal distribution (2 − 2 = 2; Figure 2e). Conversely, in bees, both GC and ENC display bimodal distributions, but their difference (GC-ENC) results in a unimodal distribution (2 − 2 = 1; Figure 2d). Similarly, in rice, both GC and ENC are bimodal, but their sum (GC+ENC) forms a unimodal distribution (2 + 2 = 1; Figure 2h). However, in mice, GC and ENC are bimodal, but their sum (GC+ENC) forms a bimodal distribution (2 + 2 = 2). These contrasting patterns suggest that GC content and codon usage bias interact differently across species, potentially reflecting underlying biological, evolutionary, and functional constraints.

### 4.2. Correlation Between GC Content and ENC Across Species

A critical insight from my study is that the correlation between GC content and ENC differs among species (Figure 3, Figure 4, Figure 5 and Figure 6). In mice and rice, these two parameters show a negative correlation, whereas in bees and wasps, they are positively correlated. This discrepancy indicates fundamental differences in codon usage optimization strategies across species. A negative correlation, as seen in mice and rice, suggests that genes with high GC content tend to use a more limited subset of codons, possibly reflecting strong selection for translational efficiency. In contrast, the positive correlation in bees implies that genes with high GC content use a broader range of codons, potentially indicating different selection pressures on codon usage.

### 4.3. Evolutionary and Functional Implications

The observed differences in GC and ENC distributions, as well as their correlation patterns, may be driven by multiple evolutionary forces. In insects such as bees, codon usage bias is strongly influenced by selection for translational efficiency, tRNA availability, and mutational biases [37,38,39,40]. The bimodal GC distribution in bees indicates that they maintain two distinct clusters of genes with differing GC content. The lower GC peak in bees, which falls below the typical GC content of most species, may indicate different selective pressures acting on subsets of the bee genome.

Bees exhibit distinct differences between their two GC peaks. The low GC peak, which can also be represented as “observed over expected” (o/e) [12], contains higher levels of DNA methylation at CpG sites compared to the high GC peak. Notably, “CpG” includes the “p” to distinguish it from GC content, which represents the proportion of guanine and cytosine in a nuclear-encoded gene. This finding is paradoxical because, despite having fewer CpG sites, the low GC peak exhibits greater DNA methylation than the high GC peak. This pattern is reminiscent of CpG islands in mammals, which are typically characterized by high GC content but relatively low levels of CpG DNA methylation (reviewed in [41,42,43]).

I observed a strong enrichment of transcription activators and *HOX* genes among the top 200 genes in the high GC peak (FDR = 10^−38^ for GO:0000981~DNA-binding transcription factor activity, RNA polymerase II-specific; FDR = 10^−25^ for IPR001356:Homeobox domain). My findings here, and in my previous publication on bee genomics, suggest that the transcriptional activation and/or alternative mRNA splicing of these genes are likely influenced by differential DNA methylation and hydroxymethylation at CpG sites in bees [12].

In plants like rice, the bimodal GC and ENC distributions reflect distinct classes of genes, potentially corresponding to different functional categories. The GC bimodal distribution was first discovered in plants in the 1980s and called “isochores” [44,45,46]. ENC values have also been shown to have bimodal distributions. For example, when ENC (effective number of codons) was plotted against the expected ENC based on GC3 content, the plant *Magnolia lotungensis*, an extremely endangered endemic tree in China, was found to have a bimodal distribution [47].

### 4.4. Implications of GC-ENC and GC+ENC Transformations

The unexpected transformation of bimodal distributions into unimodal ones (or vice versa) when GC and ENC are combined suggests that codon usage and GC content exhibit complex, non-independent relationships. In bees, where GC-ENC forms a unimodal distribution despite both components being bimodal, the subtraction operation may reveal an underlying functional relationship that aligns codon usage patterns more uniformly (Figure 3g). This could be due to selection pressures balancing codon usage bias across different genomic regions.

In mice, the unimodal GC and ENC distributions leading to a bimodal GC-ENC distribution suggest that codon usage efficiency varies among gene subsets, potentially reflecting translational optimization (Figure 4, bottom row). The emergence of a bimodal GC-ENC pattern implies that two distinct gene groups exist, possibly corresponding to highly expressed and moderately expressed genes with differing selection pressures.

For rice, the transformation of bimodal GC and ENC distributions into a unimodal GC+ENC distribution suggests that genes with different GC content and codon usage bias contribute to a broad, continuous range when considered together (Figure 2b). This may indicate that codon usage adaptation is influenced by multiple interacting factors, such as transcriptional regulation, GC-biased gene conversion, and evolutionary constraints.

### 4.5. Experimental Validation of the Possible Importance Between the Correlation Between GC Content and ENC Levels

I speculate that the correlations between GC content and ENC levels have important functional consequences that can be experimentally validated. One could test, for instance, whether transgenes in bees are more highly transcribed and/or translated when the GC content is high and the ENC levels are high, versus when GC content is high and ENC levels are low. I predict that ENC-high/GC-high genes would be more highly expressed in bees, which have a positive correlation between ENC and GC, and more poorly expressed in rice, which have a negative correlation between ENC and GC. Conversely, I predict that ENC-high/GC-low genes would be more highly expressed in rice, which have a strong positive correlation between ENC and GC. Experimental validation can be done with reporter genes, such as luciferase or GFP, or with genes that are important in biotechnology, such as Cas9 and its various derivatives. However, such experiments would be difficult because one would need to control for possible RNA secondary structures and RNA binding protein binding sites.

### 4.6. Further Uses for the Two-Rank Order Normalization (TRON) Approach

I developed the TRON mathematical approach to systematically and efficiently compare the correlations between GC content and ENC levels. These two metrics cannot be directly compared because they use different scales—GC content ranges from 0.00 to 1.00, while ENC ranges from 20 to 61, representing the number of codons in a gene. Although GC content appears normalized due to its 0 to 1 range, in reality, it typically falls between approximately 0.20 and 0.80 because some amino acids, such as methionine (ATG), always contain a G in their codon. ENC, on the other hand, can be rescaled by dividing by 61, creating a normalized range between approximately 0.33 and 1.00. While these normalization techniques yield results similar to TRON, TRON is a more straightforward and versatile method for comparing multiple variables.

Beyond GC content and ENC levels, TRON can be extended to incorporate additional gene characteristics, such as RNA secondary structure rank, GC-clamp rank [48,49], and RNA pseudoknot stability rank [50,51]. Any RNA-related feature can be integrated into TRON, making it highly adaptable. As the field of epitranscriptomics advances and features like m6A modification levels in mRNAs become better characterized, these can also be included in high-dimensional TRON analyses.

In an extended TRON framework, multiple characteristics—such as GC content, ENC levels, and m6A modifications—could be analyzed pairwise or plotted in three-dimensional space. This approach has the potential to reveal previously hidden correlations and clustering patterns among genes, offering new insights into the relationships between nucleotide composition, codon usage bias, and RNA modifications.

### 4.7. Broader Implications and Future Directions

My findings provide novel insights into the interplay between GC content and codon usage across species, revealing unexpected distributional transformations and correlation patterns. These results highlight the complexity of genome evolution and suggest that selection for codon usage optimization varies widely among taxa. Future research should investigate the mechanistic basis of these patterns, including the role of tRNA abundance, translation efficiency, and GC-biased gene conversion.

Further studies should also explore additional species to determine whether these trends hold across broader evolutionary lineages. Comparative genomics approaches integrating transcriptome and proteome data could provide deeper insights into how GC content and codon usage influence gene expression and translational efficiency. Moreover, experimental validation using synthetic gene constructs with varying GC content and codon usage biases could help elucidate the functional consequences of these patterns.

One consideration that is not discussed in this paper is the role of RNA secondary and tertiary structures, which are predicted to increase as GC content increases, given the higher melting temperature of GC base pairs compared with AT base pairs. RNA secondary structure is important in the design of RNA viruses, for instance, because double-stranded RNA is more stable than single-stranded RNA. Also, the double-stranded nature of RNA can have both positive and negative effects on translation efficiency of genes [52].

Finally, I did not discuss possible evolutionary reasons for why GC vs ENC correlations increase in low GC-content species such as bees and decrease in high GC-content species such as rice. It could be trivial in that increasing GC content in nuclear-encoded genes in AT-rich species provides more GC codons to use and thereby increases the ENC levels for those genes. Conversely, increasing GC content in nuclear-encoded genes in GC-rich species provides fewer AT codons to use, and thereby decreases the ENC levels for those genes. Even if the explanation for the correlations between GC and ENC are trivial, there could be important emergent properties that occur, such as high GC content having more stable secondary structures.

The correlations that I identified could also be profound and lead to new understandings of evolutionary processes. The evolutionary conservation of olfactory genes in low GC and high ENC regions could reflect their need to be rapidly translated in olfactory neurons, for instance. Similarly, the evolutionary conservation of *HOX* genes in high GC and low ENC regions, which I observed in both insects and mammals, could reflect their possible complex RNA secondary structures needed for proper translation or RNA modifications [53,54], for instance.

## 5. Conclusions

In conclusion, my study uncovers intriguing species-specific relationships between GC content and codon usage bias, demonstrating that simple numerical transformations can reveal underlying biological constraints. The unexpected distributional patterns across mice, bees, and rice highlight the complexity of genome evolution and codon usage adaptation. These findings emphasize the importance of considering multiple evolutionary and functional factors when interpreting codon usage bias and suggest promising avenues for future research in comparative genomics and translational regulation.

## Figures and Tables

**Figure 6 genes-16-00432-f006:**
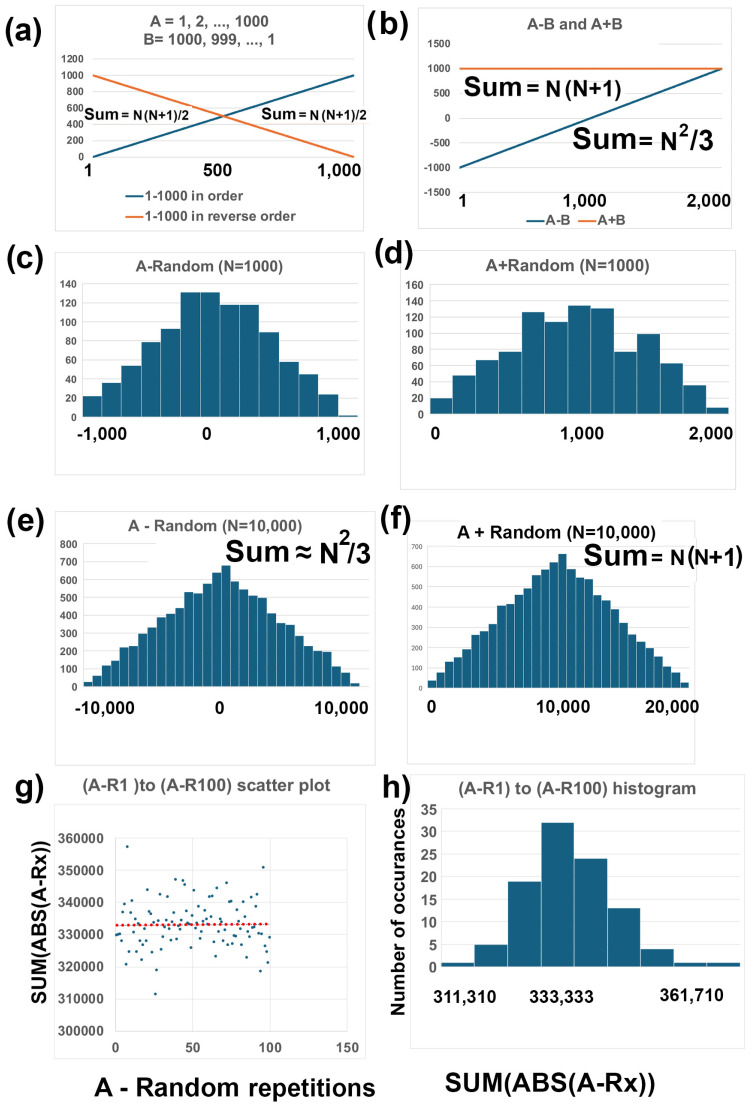
Combinatorial effects of adding or subtracting GC and ENC ranks. (**a**) Line A (1, 2, …, 1000) (red) and Line B (1000, 999, …, 1) are plotted. Column A on Excel™ has the numbers for Line A and column B has the numbers for Line B. (**b**) Line A minus Line B (A−B) (blue) and Line A+B (red) are plotted. A−B was made by selecting column A (rows 1–1000) and subtracting column B (rows 1–1000) and placing the results in column C. A+B was made by selecting column 1 and adding column 2 and placing the results in column D. (**c**) A histogram of Line A minus a randomization of Line A (Random) is plotted (A-Random). Random was generated on Excel™ with the RANDARRAY function, i.e., =SORTBY(A_1_:A_1000_,RANDARRAY(1000)). The results were placed in column E. The histogram was made by selecting column E (rows 1–1000) and selecting the histogram chart under the INSERT tab. (**d**) A histogram of Line A plus a randomization of Line A (R) is plotted (A+Random). The results of A+Random was inserted into column F. (**e**) A histogram of Line A’ (1, 2, …, 10,000) (column G) minus a randomization of A’ (column H) and placed in column I (A’-Random’). The steps in C were repeated using numbers 1–10,000 for line A’ and randomization of numbers 1–10,000 for Random’. The area was determined by the equation (SUM(ABS(I_1_:I_10,000_). ABS (absolute value) was used in this equation because half of the numbers are negative. The area can also be approximated as N^2^/3, where N is the number of rows, in this case there are 10,000 rows (see methods). (**f**) A histogram of Line A’ plus Random’ and placed in column J (A’+Random’). The area was determined by the equation (SUM(J_1_:J_10,000_)) = N(N + 1)/2. (**g**) A scatter plot of 100 repetitions of SUM(ABS(A_1_:A_1000_) − (R_1_:R_1000_)), where R is a randomization of the numbers between 1 and 1000 using the equation SORTBY(A_1_:A_1000_,RANDARRAY(1000). The red line shows the average = 333,023 +/− 7360, which is equivalent to N^2^/3 +/− 2%. (**h**) A histogram of the results in g, where the x-axis is SUM(ABS(A_1_:A_1000_) − (R_1_:R_1000_)) and the y-axis is the number of times that range of number occurred in 100 repetitions.

**Figure 7 genes-16-00432-f007:**
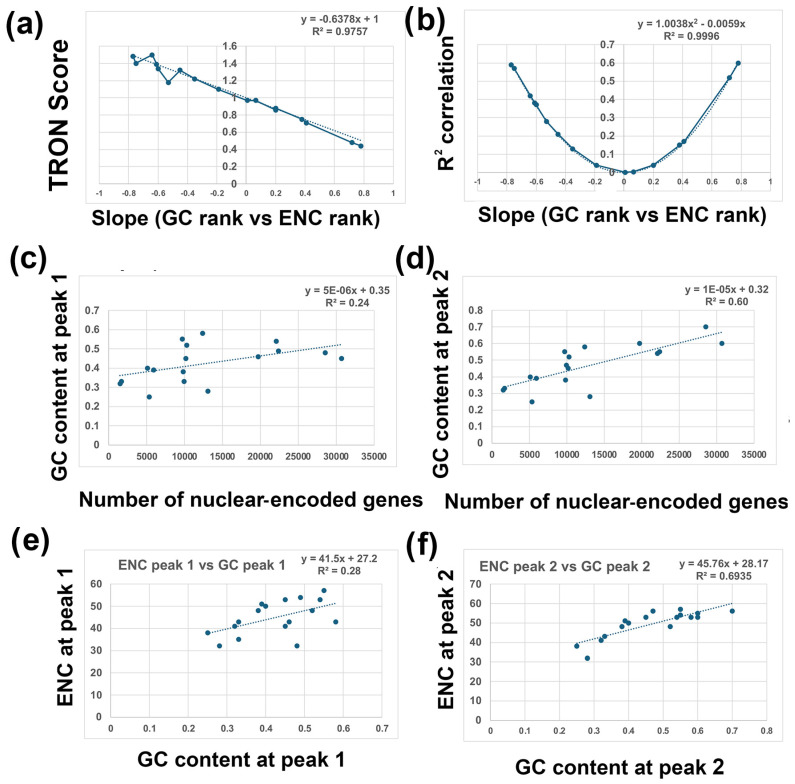
Correlations between GC content, ENC, and the number of nuclear-encoded genes. Data for all graphs is from Table 1. (**a**) Plot of TRON score (y-axis) versus slope (GC rank vs. ENC rank) (x-axis) for all 17 species. Species with negative slopes between GC ranks and ENC ranks are on the left and species with positive slopes are on the right. The trendline and R-squared value is shown. TRON score is SUM(ABS((GC_1_:GC_N_) − (ENC_1_:ENC_N_))/(N^2^/3). (**b**) Plot of R-squared correlation (y-axis) versus slope (GC rank vs ENC rank) (x-axis) for all 17 species. Species with negative slopes between GC ranks and ENC ranks are on the left and species with positive slopes are on the right. The polynomial trendline and R-squared value is shown. (**c**) GC content at peak 1 (y-axis) versus number of nuclear-encoded genes (x-axis) for all 17 species. The trendline and R-squared value is shown. (**d**) GC content at peak 2 (y-axis) versus number of nuclear-encoded genes (x-axis) for all 17 species. The trendline and R-squared value is shown. (**e**) ENC level at peak 1 (y-axis) versus GC content at peak 1 (x-axis) for all 17 species. The trendline and R-squared value is shown. (**f**) ENC level at peak 2 (y-axis) versus GC content at peak 1 (x-axis) for all 17 species. The trendline and R-squared value is shown.

**Table 1 genes-16-00432-t001:** Species (17) analyzed in this study. Shown are the number of genes (N), the GC content at peaks 1 and 2, the ENC levels at peaks 1 and 2, the slope equation of the GC rank versus ENC ranks, the slopes from this equation (negative for rice and positive for bees), the R^2^ values for the slopes, the Two-Rank Order Normalizations (TRON) for GC-ENC (i.e., SUM(ABS(GC-ENC)/(N^2^/3), and the Two-Rank Order Normalization (TRON) for GC+ENC (i.e., SUM(GC + ENC)/N(N+1) = 1.00).

Common Name	Species	Genes (N)	GC Peak 1	GC Peak 2	ENC Peak 1	ENC Peak 2	Line Equation	Slope	R^2^	(GC-ENC) /(N^2^/3)	(GC+ENC) /N(N+1)
Rice	*Oryza sativa*	28,571	0.48	0.7	32	56	y = −0.77x + 25253	−0.77	0.59	1.48	1
Mosquito	*Anopheles gambiae*	12,402	0.58	0.58	43	53	y = −0.75x + 10878	−0.75	0.57	1.4	1
Puffer fish	*Takifugu rubripes*	22,107	0.54	0.54	53	53	y = −0.64x + 18180	−0.64	0.42	1.5	1
Humans	*Homo sapiens*	19,708	0.46	0.6	43	53	y = −0.61x + 15954	−0.61	0.38	1.39	1
Bread mold	*Neurospora crassa*	9728	0.55	0.55	57	57	y = −0.60x + 7830	−0.60	0.37	1.34	1
Banana	*Musa acuminata*	30,700	0.45	0.6	41	55	y = −0.53x + 23495	−0.53	0.28	1.18	1
Mouse	*Mus musculus*	22,405	0.49	0.55	54	54	y = −0.45x + 16307	−0.45	0.21	1.32	1
*E. coli* bacteria	*Escherichia coli*	10,276	0.52	0.52	48	48	y = −0.35x + 6975	−0.35	0.13	1.22	1
Pombe yeast	*pombe*	5110	0.4	0.4	50	50	y = −0.19x + 3063	−0.19	0.04	1.1	1
Methanobacteria	*Methanococcus aeolicus*	1485	0.32	0.32	41	41	y = 0.064x + 696	0.064	0.004	0.97	1
Bakers yeast	*Saccharomyces*	5958	0.39	0.39	51	51	y = 0.0064x + 2983	0.0064	0.0008	0.97	1
Honey bee	*Apis mellifera*	9918	0.33	0.47	35	56	y = 0.78x + 1103	0.78	0.6	0.44	0.99
Red paper wasp	*Polistes canadensis*	9854	0.38	0.38	48	48	y = 0.72x + 1367	0.72	0.52	0.48	0.99
Spotted fever parasite	*Rickettsia hoogstraalii*	1663	0.33	0.33	43	43	y = 0.41x + 489	0.41	0.17	0.71	1
Slime mold	*Dictyostelium discoideum*	13,078	0.28	0.28	32	32	y = 0.38x + 4044	0.38	0.15	0.75	1
Mustard weed	*Arabidopsis thaliana*	10,160	0.45	0.45	53	53	y = 0.20x + 4066	0.20	0.04	0.86	1
Malaria parasite	*Plasmodium falciparum*	5321	0.25	0.25	38	38	y = 0.20x + 2141	0.20	0.04	0.88	1

## Data Availability

All of the GC and ENC data used in this paper are from the codon statistics database (https://codonstatsdb.unr.edu, accessed on 15 March 2025) [31]. Data analyses for many species was done using Microsoft Excel and the data analyses are available upon request.

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
