# Peer review of "GC Content in Nuclear-Encoded Genes and Effective Number of Codons (ENC) Are Positively Correlated in AT-Rich Species and Negatively Correlated in GC-Rich Species"

_genes, 2025, doi:10.3390/genes16040432_

Round 1
Reviewer 1 Report
Comments and Suggestions for Authors
Dear Authors,
Reviewer comments genes-3527159
The manuscript entitled „Unexpected relationships between GC content and effective number of codons (ENC) in various species“ represents an interesting bioinformatics study aimed at an investigation of the relationship between GC content and effective number of codons (ENC) in three representative species – honeybee (Apis mellifera), mouse (Mus musculus), and rice (Oryza sativa). The bioinformatics study has uncovered distinct patterns and relationships between GC content and effective number of codons (ENC) across different biological species. the manuscript has originality lying in a comparison of three diverse biological species. I can only suggest to include more species in a comparison, e.g., Arabidopsis thaliana as a model dicot angiosperm species or some unicellular eukaryotes such as yeast, or Procaryota such as cyanobacteria or Escherichia coli. The current selection of just three species including honeybee, mouse, and rice is not justified in the manuscript. Why just these species were selected for the study and not the others?
Major comment: I miss proper justification of the selected three species employed in the present study.
Otherwise, I have only a few formal comments on the manuscript which are provdied below:
1/ Please correct the sceinfic name for mouse, i.e., „Mus musculus“, not „Mus musculis“.
2/ In the statements starting with the words „This suggests…“ I recommend the author to add some useful noun, e.g., „This finding suggests…“ or „This result suggests…“
Final recommendation: Accept after a minor revision.

Reviewer 2 Report
Comments and Suggestions for Authors
The manuscript explores the unexpected relationships between GC content and the effective number of codons (ENC) across Apis mellifera, Mus musculus, and Oryza sativa, uncovering species-specific correlations that shape codon usage patterns. While honeybees show a positive correlation between GC content and ENC, mice and rice exhibit a negative one. These differences lead to surprising distributional shifts: subtracting ENC from GC rankings in honeybees transforms bimodal distributions into unimodal ones, while adding the two in rice achieves a similar effect. The study suggests that codon usage constraints are not uniform across taxa and may be influenced by evolutionary selection, translation efficiency, and epigenetic factors.
The comparative approach provides valuable insights into how genomic composition affects gene expression, and the mathematical modeling effectively clarifies the observed trends. The findings could have practical applications in synthetic biology, particularly in gene design strategies where codon optimization plays a role. Overall, the research presents a compelling analysis of codon usage variation, offering fresh perspectives on the interplay between genomic composition and gene regulation.
Some suggestions to improve the manuscript:
- Explain why GC-ENC relationships differ across species and their functional implications.
- Briefly explain why honeybee, mouse, and rice were chosen for the study.
- Address potential limitations of using ranked data instead of raw values.
- Improve figure clarity, especially for mathematical transformations.
- Provide a concrete example of how these findings could be used in synthetic biology.
- Mention functional assays to test GC-ENC effects on translation efficiency but avoid presenting it as a major limitation.
- Simplify dense mathematical explanations and define technical terms for broader accessibility.
Reviewer 3 Report
Comments and Suggestions for Authors
This is an interesting manuscript that examines the DNA GC base content as a function of the number of codons used in three different species. The author finds that there is not a uniform match of GC content to codon usage (Figure 1). Given the effort to make these three comparisons, it is left to the reader to imagine how this might look when 20 different species might be compared. The major limitation is the interpretation as to what this means. Major complicating features might be the number of different cell types in the organism and whether the organism is a homotherm or not. In short, it is not possible to draw any direct conclusion based upon these studies.
Concerns
- Title – it is not clear why the relationships observed are “unexpected”. This would necessitate knowing what the “obvious expected” findings should be. This has not been made clear.
- In multiple places the author uses “we”. Given that the author did all of the work, “I” would be more appropriate.
- Figure 1 – it is not clear if the ENC value is a relative number or yes/no the codon is present. Additionally, the numbering for the x axis is not clear (i.e. should the scale go from 0 to 100% GC content?).
- Line 134 – “Given that both the GC and ENC distributions are unimodal, one might expect …”. It is not clear why one might expect this. Line 145 - “This unexpected result (i.e. 1 – 1 = 2) suggests …”. It is not clear why this is unexpected.
- Line 230 – “This suggests that genes …”. This is pure speculation.
In sum, this reviewer is unable to understand what the relationships described here (GC content and codon usage) have to do with the biology of the organisms examined.
Reviewer 4 Report
Comments and Suggestions for Authors
The author describes that ENC and GC rankings were positively correlated in honeybees and negatively correlated in mice and rice. In honeybees, subtracting ENC rank from GC rank (GC - ENC) transformed bimodal distributions into a unimodal one. In rice, adding ENC and GC (GC + ENC) had the same effect. These results suggest that codon usage constraints vary across species. However, I would like to ask for the author to answer to my comments described below, because I consider that the author might misunderstand about the analysis of genetic data.
Major comments
 Three genome data of mouse, rice and honeybee, of which GC contents of genes are characteristic respectively, were analyzed in the manuscript. For example, GC content distribution of mouse is umimodal, although the distribution is slightly expanding to a higher GC content. On the contrary, GC content distributions of rice and honeybee are bimodal. Two peaks are observed around 50% and extremely GC-rich region in the distribution of genes encoded by the rice genome. In the case of honeybee genome, two peaks are detected around 50% and extremely AT-rich region. It is easily supposed that the usages of AU-rich codons is restricted in GC-rich genes and the usage of GC-rich codons is restricted in AT-rich genes. Therefore, probability using 61 codons should increase in genes having around 50% GC contents and the probability should decrease in extremely GC-rich and AT-rich genes.
 The results given in the manuscript are interpreted based on the considerations described above.
- ENC(vertical axis) and GC (horizontal axis) rankings are positively correlated in honeybees (Figure 1 (c)) and negatively correlated in mouse (Figure 1 (a)) and rice (Figure 1 (b)). The reasons are because moderately GC-rich genes and extremely GC-rich genes are contained in the genomes of mouse and rice, respectively and extremely AT-rich genes are contained in honeybee genome.
- The results shown in Figure 2 (a) can be interpreted as follows. The reason why mouse genes incline towards a high ENC side is because many mouse genes distributes around 50% GC content, which causes the usage of an extensive range of codons and, in parallel, a small number of moderately GC-rich genes and AT-rich genes, which are contained in mouse genome, cause tailing of ENC pattern towards AT-rich side.
- The reason, why the result of ENC of rice is bimodal (Figure 2 (b)), is because ENC becomes low in extremely GC-rich genes and ENC of genes having around 50% GC contents becomes high. This can be confirmed by the fact that ENC of genes having around 50% GC content forms the large peak (more than 50 ENC) and ENC of genes having extremely high GC contents (around 70%) forms a smaller peak (around 30~40 ENC).
- It can be also confirmed by the results of honeybee genome. That is, the large peak formed by AT-rich genes (around 30% GC content) corresponds to the large peak formed by ENC below 45 (Figure 2 (c)). The smaller peak composed of honeybee genes having around 50% GC contents corresponds to the smaller peak of more than ENC 50 (Figure 2 (c)).
- The problem of the manuscript is that it seems to me that the results, which were obtained by subtracting or adding the GC-content distribution and ENC distribution after ranking them, are meaningless, because the two distributions are unrelated each other.
 Of course, many wrong interpretations may be contained in my comments above. Anyway, I would like to receive the authors replies to my comments described.
Round 2
Reviewer 3 Report
Comments and Suggestions for Authors
This is a much improved manuscript especially with the addition of 14 more species which makes this more comprehensive. In particular, it lends much more support to the title of the manuscript. However, this reviewer is still left with the question “why”. The only simple explanation could be the strength of the base pairing of the mRNA with the aminoacyl-tRNA with the understanding that AT base pairs are less stable than GC base pairs. In this case, for AT rich genomes, which would normally have more AT rich base pairings in the codon-anticodon interactions, it might be necessary to invoke the use of an increased number of GC base pairings to avoid mistakes in translation and thus AT rich genomes might be expected to have increased ENC values. But like many of the ideas put forth in the discussion, while perhaps reasonable, it is speculative.
Concerns
- As noted above, there is the bigger question of “why”. Does GC content of the genome reflect environment of the organism (warm vs. cold) or body temperature (homotherm) or something else?
- The Figure legend to Figure 1 provides sufficient information for interpretation. Many of the other figures do not. For example, it is not clear what the value is that comes from either GC rank – ENC rank vs. GC rank + ENC rank. By that same token, the value of figures 7 and 8 is not obvious.
- The discussion is filled with speculation ( (from lines 1360 to 1414). It is not clear that this is warranted. It would be much for useful to comments that might directly lead to experimental evaluation.
Minor comments
- The author still has a number of “we” in the manuscript which should be I (see especially lines 829 to 843, although there are others).
- Title – AT rich species should be AT Rich Species.
- Line 124 – should be “such as GC content and ENC levels”.
- Line 252 – “This equation is name this called this because” – This needs to be reworded.
- Line 1458 – Should be “One consideration that is not discussed…”
Author Response
Concerns
- As noted above, there is the bigger question of “why”. Does GC content of the genome reflect environment of the organism (warm vs. cold) or body temperature (homotherm) or something else?
I added examples of "why" to several parts of the paper. E.g., adaptation to extreme environments as you suggested.
- The Figure legend to Figure 1 provides sufficient information for interpretation. Many of the other figures do not. For example, it is not clear what the value is that comes from either GC rank – ENC rank vs. GC rank + ENC rank. By that same token, the value of figures 7 and 8 is not obvious.
I deleted Figure 8 (GO analyses). However, Figure 7 a is making the most important point in the paper. I emphasize this point in the new Graphic abstract.
- The discussion is filled with speculation ( (from lines 1360 to 1414). It is not clear that this is warranted. It would be much for useful to comments that might directly lead to experimental evaluation.
I deleted most of these paragraphs and added an experimental validation section. I agree it is needed.
Minor comments these have all been addressed in vs3.
- The author still has a number of “we” in the manuscript which should be I (see especially lines 829 to 843, although there are others).
- Title – AT rich species should be AT Rich Species.
- Line 124 – should be “such as GC content and ENC levels”.
- Line 252 – “This equation is name this called this because” – This needs to be reworded.
- Line 1458 – Should be “One consideration that is not discussed…”
Reviewer 4 Report
Comments and Suggestions for Authors
The author have added explanations replying to my comments in the revised manuscripts. So, I would like to recommend the editor to accept the revised manuscript.
Minor comment
“4.6. Broader Implications and Future Directions” should be “4.7. Broader Implications and Future Directions”.
Author Response
I appreciate your help in making this a more interesting paper.
Round 3
Reviewer 3 Report
Comments and Suggestions for Authors
While this reviewer will continue to wonder "why", the revised manuscript is now acceptable. The concern about the amount of speculation is confirmed by the author with useful future experiments that might shed light on his observations. Note, in the title it should be AT Rich Species, not AT Rich species.